# Gadolinium Accumulation and Toxicity on In Vitro Grown *Stevia rebaudiana:* A Case-Study on Gadobutrol

**DOI:** 10.3390/ijms231911368

**Published:** 2022-09-26

**Authors:** Violeta Florina Scurtu, Doina Clapa, Loredana Florina Leopold, Floricuța Ranga, Ștefania D. Iancu, Adrian Ionuț Cadiș, Vasile Coman, Sonia Ancuța Socaci, Augustin C. Moț, Cristina Coman

**Affiliations:** 1Life Sciences Institute, University of Agricultural Sciences and Veterinary Medicine, 3-5 Calea Mănăștur, 400372 Cluj-Napoca, Romania; 2Faculty of Chemistry and Chemical Engineering, Babeș-Bolyai University, Arany János 11, 400028 Cluj-Napoca, Romania; 3Faculty of Food Science and Technology, University of Agricultural Sciences and Veterinary Medicine, 3-5 Calea Mănăștur, 400372 Cluj-Napoca, Romania; 4Faculty of Physics, Babeș-Bolyai University, 1 Kogalniceanu, 400084 Cluj-Napoca, Romania; 5Raluca Ripan Institute for Research in Chemistry, Babeș-Bolyai University, 30 Fântânele, 400294 Cluj-Napoca, Romania

**Keywords:** gadobutrol, toxicity, free radicals, accumulation, growth, carotenoids, chlorophylls

## Abstract

Gadolinium-based contrast agents are molecular complexes which are extensively used for diagnostic purposes. Apart from their tremendous contribution to disease diagnostics, there are several issues related to their use. They are extremely stable complexes and potential contaminants of surface and ground waters, an issue which is documented worldwide. The irrigation of fields with contaminated surface waters or their fertilization with sludge from wastewater treatment plants can lead to the introduction of Gd into the human food supply chain. Thus, this study focused on the potential toxicity of Gd on plants. For this purpose, we have studied the molecular effects of gadobutrol (a well-known MRI contrast agent) exposure on in vitro-grown *Stevia rebaudiana*. The effects of gadobutrol on plant morphology, on relevant plant metabolites such as chlorophylls, carotenoids, ascorbic acids (HPLC), minerals (ICP-OES), and on the generation of free radical species (MDA assay and EPR) were assessed. Exposures of 0.01, 0.05, 0.1, 1, and 3 mM gadobutrol were used. We found a correlation between the gadobutrol dose and the plant growth and concentration of metabolites. Above the 0.1. mM dose of gadobutrol, the toxic effects of Gd^+3^ ions became significant.

## 1. Introduction

Gadolinium-based contrast agents (GBCA) are molecular complexes containing Gd as the central metal. Since their approval as contrast agents in magnetic resonance imaging (MRI) in 1980s [1,2], these complexes have become indispensable for diagnostic imaging. They are currently being used in up to 50% of total MRI investigations.

There are some concerns related to the long-term effects of GBCA on the human health of patients undergoing diagnosis tests. While long considered to be safe for use, recent studies have indicated that administration GBCA results in retention of small amounts of Gd in brain tissues [3,4,5,6]. Moreover, the Gd originating from linear complexes is more likely to be accumulated than the one from macrocyclic GBCA [4,6]. Based on these issues, the European Medicines Agency (EMA) started a review in 2016 [7] regarding the risk of Gd deposition in brain tissues as a result of the use of contrast agents in patients that undergo MRI scans. Following the review, even if no evidence was found that Gd deposition in the brain would cause any harm to patients, EMA has recommended restrictions [8] regarding the use of GBCA. Thus, EMA suspended the use of some linear complexes and limited the use of others in cases where they were crucial for diagnostic needs. They retained all macrocyclic complexes already in use.

Apart from this, there are also environmental issues related to the use of GBCA. After administration to patients, GBCA are mainly excreted unmetabolized by the kidneys, leading to a major percentage of all GBCA being present in hospital effluents and further reaching the waste-water treatment plants (WWTPs) [2,9]. Because of their very high stability and high water solubility, they are impossible to remove in WWTPs. Previous studies have shown that less than 10% of the GBCA could be removed during water treatment procedures in WWTPs [10]. In consequence, there is a real possibility that these complexes reach surface and ground waters, as well as rivers and lakes.

A study performed on WWTPs in Switzerland estimated that the annual discharge of Gd to surface waters was 90 kg of from every single WWTP [11]. The presence of anthropogenic Gd (Gd which is the result of human activity) has been documented both in surface and ground waters [9], but also in German rivers and tap water in Berlin and Dusseldorf [12], and tap-water-based beverages such as Coca Cola from fast-food franchises (McDonalds, Burger King) in six major cities in Germany [13]. Positive values of anthropogenic Gd were found not only found in Germany, but in natural waters from other European countries (e.g., Italy, France, Nordic countries, and the Czech Republic) [14,15,16,17], but also in North and South America [18], Australia [19], Asia [12]. Basically, the waters from the industrialized, densely populated areas are the most susceptible to anthropogenic Gd contamination. A study from 2009 showed that after passing through Prague, the Gd concentration in the Valtra river increased from 4.5 to 19.2 nmol/m^3^ [20].

Another matter of concern is related to the increases in anthropogenic Gd concentrations over time. Tepe et al. (2014) found that anthropogenic Gd levels in Berlin tap water have increased by 1.5 and 11.5-fold in just three years (depending on the district), between 2009–2012 [12]. Hatje et al. (2016) performed a 20-year study to assess Gd concentration in surface waters collected in a transect of San Francisco Bay and found increases in the Gd concentration from 8.27 to 112 pmol/kg from 1993 to 2013 [21].

Kümmerer and Helmers (2000) performed theoretical and experimental studies via ICP-MS and found that emissions of Gd^+3^ ions from Freiburg University Hospital in Germany was in the range 2.1–4.5 kg/year, yielding theoretical concentrations of 8.5–30.1 μg/L in the hospital effluent [22]. The total estimated annual emissions from German hospitals are between 484–1160 kg/year [22].

Apart from their direct entrance pathway in the human body subjected to MRI scans, GBCA can also reach the human body indirectly, from the food chain. Possible entrance routes of the GBCA in the food chain involve: (i) from plants grown on fields which are irrigated with contaminated surface waters, or which are fertilized with sludge from the WWTPs; (ii) from contaminated tap water which is directly consumed by humans or used for preparation of other foods or beverages; and (iii) from animals which drink contaminated water.

Because of the real possibility of these Gd complexes reaching the human food chain from plants exposed to anthropogenic Gd, the main goal of the present work was to focus on the potential toxicity of Gd complexes on plants. More precisely, we have assessed the effects of gadobutrol (GB), a widely used macrocyclic contrast agent, on *Stevia rebaudiana* plants grown in vitro. Stevia cultivation has increased in importance in the food industry in recent years due to its high content of steviol glycosides used as sweeteners. Steviol glycosides from Stevia represent an suitable alternative for sugar and artificial sweeteners, being 250–300 times sweeter than sugar and, at the same time, a low-calorie sweetener [23].

The potential toxicity that Gd^3+^ exposure exerts on in vitro grown *S. rebaudiana* was assessed by studying the effects on plant growth and plant biomass, by quantifying the concentrations of relevant plant metabolites such as chlorophylls, carotenoids, ascorbic acids, and by evaluating the possibility of generation of free radical species (MDA assay and EPR).

To date, there are studies proving the toxicity of Gd on aquatic ecosystems such as algae, microcrustaceans, and mussels [24,25,26]. Some studies on the toxicity of Gd on plants such as rice, maize, tomato, and *Arabidopsis thaliana* have been published [27,28,29,30,31,32], but there are still many questions to be addressed.

## 2. Results

### 2.1. Plant Growth and Morphology

The influence of gadolinium on the morphology of *S. rebaudiana* was assessed by quantifying the shoot and root length and the dry weight plant biomass after 28 days of growth in media containing 0.01, 0.05, 0.1, 1, and 3 mM GB. The plants treated with GB were compared to control samples plants grown without exposure to Gd. As can be seen in Figure 1, significant differences in the plant morphology were associated with the high concentration treatments. The growth of plants exposed to 1 mM and 3 mM gadobutrol was severely inhibited. Looking at the shoot length (Figure 1b), it could be seen that it diminished with increasing the Gd dose. For the 0.01, 1, and 3 mM treatments, the shoot length was decreased by 31.57%, 45%, 79.73%, respectively. For the highest exposure of 3 mM, the root length (Figure 1c) was decreased by 85.94% compared to controls, and for many plants grown under these conditions, the roots were completely missing. The biomass production (Figure 1d) was significantly reduced by 23.87% (0.1 mM), 41.77% (1 mM), and 42.33% (3 mM).

### 2.2. Antioxidant Molecules and Photosynthetic Pigments

Apart from morphological changes, the influence of GB on the synthesis of different plant metabolites was also investigated. Variations in the concentrations of ascorbic (AA) and dehydroascorbic acid molecules (DHAA), as well as relevant yellow (lutein, zeaxanthin, β-carotene) and green (chlorophylls A and B) photosynthetic pigments were assessed.

Increased concentrations of AA were observed upon Gd exposure (Figure 2), with highest levels observed for the 0.1 and 1 mM doses (an 89.63% and 82% increase compared to controls), while the 3 mM dose induced an increase of only 10%. The DHAA levels showed a tendency of decrease with the increase in the gadobutrol dose. For the highest doses of 1 and 3 mM, the DHAA values dropped by 21.83% and 28.93% compared to controls, but the changes were not so intense as for AA.

The impact of Gd on carotenoids and chlorophylls synthesis was dose dependent. The variations in the concentrations of lutein, zeaxanthin, and beta-carotene showed a similar trend (Figure 3). On one hand, a dose-dependent increase was measured up to 0.1 mM exposures. On the other hand, high GB doses, of 1 and 3 mM significantly decreased carotenoid production. Compared to controls, lutein production decreased by 29.38% (1 mM) and by 38.16% (3 mM), zeaxanthin decreased by 28.55% (1 mM) and 54.34% (3 mM) and beta-carotene decreased by 35.25% (1 mM) and 40.12% (3 mM).

The levels of the chlorophylls A and B were also compared across different treatments. The concentrations of chlorophylls basically followed the same trend as the carotenoids, as can be seen in Table 1. Treatments up to 0.1 mM GB led to increases in chlorophyll A and B production. For the 0.1 mM treatment, the rise was by 52% (chlorophyll A) and by 12.7% (chlorophyll B). At the highest doses of 1 and 3 mM, the chlorophyll content decreased significantly. For the highest dose of 3 mM, the chlorophylls A and B production was almost halved compared to controls (decreased by 52.76% for Chl A and 58.92% for Chl B).

Regarding the chl A/B ratio, this increased from 2.23 (control) to 3.01 (0.1 mM) and 2.56 (3 mM).

### 2.3. Gd Accumulation and Nutrient Levels

#### 2.3.1. Gd Uptake in the Plant Tissues

Gd ions were internalized by the *S. rebaudiana* plant in a dose-dependent manner. This was shown by ICP-OES results, which proved there was an increase in Gd ions concentrations in plant tissue samples with the increase in the GB exposure (Figure 4). The analyses were carried out on the whole plant. For the 0.01, 0.05, 0.1, 1, and 3 mM treatments, the Gd ion concentrations in the plants were 7.88, 39.61, 86.8, 645, and 744 µg/g DW, respectively. Similar results were found by Liu et al. [31] for Gd uptake on A. thaliana, in a study on low-concentration Gd exposures, up to 0.2 mM.

#### 2.3.2. Gd Influence on Nutrient Uptake

Except for Na content—which increased significantly up to 0.1 mM exposure, but then reached the same values as the control for the 3 mM sample—all the other minerals showed strong changes for the highest exposure of GB (3 mM) when compared to controls. The Cu was not detected at this exposure; the levels of Ca, Fe, K, Mn, Zn dropped by 33.34% (Ca), 29.14% (Fe), 21.81% (K), 37.25% (Mn), and 55% (Zn). The levels of Mg showed increases of 59.61% compared to controls for this highest exposure.

Among the analyzed minerals, the levels of Ca, Mg, and Zn were only moderately affected by lower exposures. Fe and Mn levels were not affected up to 0.1 mM GB, while Ca was very little influenced up to 1 mM.

Strongest effects were exerted on the content of Zn, Na, K, Mg, Cu, even from the lowest exposures. The Cu, which was initially present in controls at 0.011 µg/g DW became undetectable for the 1 mM and 3 mM exposures. Zn levels dropped from the lowest doses of GB (0.01 mM) by 47.19%, reaching 55% for the highest dose of 3 mM GB. Na levels increased with respect to controls up to 1 mM, reaching maximum concentration for 0.1 mM exposure (increasing by 75.51%), then dropped at 3 mM as discussed previously. K levels also increased up to 1 mM, with a maximum (by 26.27%) at 0.1 mM and a drop at 3 mM. Mg ions concentrations increased dose dependently, with the maximum 59.61% increase being reached for the 3 mM treatment.

Regarding Cd, Pb, Cr, they were not present in the samples or were detected only in traces.

Nutrient levels within other plants such as maize and rice were also found to vary in response to Gd treatments. In a study on maize [32] (hydroponics, 0, 0.63, 6.3, 63 µM Gd treatments), K, Fe, Zn, Cu contents did not suffer significant changes, while Ca, Mg, and Mn contents in the roots were influenced by the high Gd dose. The study also reported reduction in shoot growth (by 67%) and root biomass (by 35%) for the highest dose of 63 µM Gd. In a study on rice seedlings (hydroponics, 0–1 mM treatments) Zhang et al. [28] found that in rice shoots, the content of Ca, Mg, and Zn decreased significantly at the high doses (0.1, 1 mM). K, Na, Zn were mainly affected at the highest dose (1 mM), while Fe and Cu showed significant inhibition already from the lowest doses. Regarding the roots, no significant difference was found in Mg content, while K, Ca, Cu, Na, and Zn had significant inhibition at the highest doses. The concentrations of Fe and Mn in the roots increased significantly already from the lowest doses of Gd.

### 2.4. MDA and EPR

MDA is one very important marker of oxidative damage. It is formed as final product of fatty acids peroxidation in cells. An increase in free radicals causes increased levels of MDA.

To be able to quantify the levels of oxidative stress, we aimed at quantifying the amount of malondialdehyde (MDA), spectrophotometrically, by the MDA-TBA complex that gives absorption maxima around 530 nm. The assay led to the formation of a yellow complex with characteristic UV-Vis spectra that showed two absorption bands around 450 and 530 nm, respectively.

According to previous studies, the absorption peak at 530 nm is caused by the MDA–TBA complex formed by the oxidation of polyunsaturated fatty acids, while the peak with absorption maxima at 450 nm is caused by oxidation of mono-unsaturated fatty acids [33]. In the present case, little difference was observed in the absorbance of the MDA-TBA peak at 530 nm, but the peak at 450 nm increased in intensity with the increase in the GB dose, the increase being most pronounced for the 3 mM treatment, reaching >three times higher values compared to controls for the 3 mM dose. For the lower doses, the increase in the 450 nm peak reached 34%. These evolutions are illustrated in Figure 5 and indicate oxidative damage at high Gd doses.

Figure 6 shows the EPR spectra of the analyzed Stevia powdered samples at room temperature. The g = 4.30 signal was specific for high spin ferric iron—probably the most common type of iron—and appeared not be dependent upon the Gd dose, except for the highest concentration, in good agreement with the ICP-OES analysis (Table 2). The very broad paramagnetic Gd signal was centered at g = 1.998 and became clearly visible only in samples originating from plants grown in high Gd content (above 0.1 mM Gd). However, estimating Gd spin concentration directly in samples was difficult due to signal overlapping with the intense Mn signal. The six highly specific hyperfine lines of the Mn centers did not indicate any modification, both qualitatively and quantitatively, for samples of the plants grown up to 0.1 mM Gd. However, plants grown at a higher Gd concentration exhibited powerful distorted Mn centers and lower Mn content, in a dose-dependent manner, indicating vital dysfunctionalities as Gd content increased, as also supported by the elemental analysis (Table 2).

Interestingly, the intensity of the EPR-detected radical in the powdered samples exhibited a distinct and well-defined U-shape curve, as shown in Figure 6b. Such a shape indicated two antagonistic processes, i.e., a beneficial impact at low Gd content and a detrimental effect at higher Gd content. Similar patterns have been observed for other constituents, such as ascorbic acid (Figure 2), carotenoids (Figure 3) and some metals (Table 2). Moreover, significantly negative correlation between radical intensity and ascorbic acid (r = −0.663, *p* < 0.05), K (r = −0.517, *p* < 0.05), Na (r = −0.812, *p* < 0.05) or positive correlation with Zn (r = 0.845, *p* < 0.05) have been observed. Therefore, an increase in ascorbic acid concentration at lower Gd content might explain the decrease in the radical intensity due to its antioxidant effect, and thus its beneficial impact. On the other hand, low Gd concentration stimulated the absorption of K and Na up to 40% which might be favorable to plant development, as shown by others [34]. Similar to the radical intensity behavior, the sharp decrease in Zn content in plants exposed to Gd, even at low concentration, is expected to influence the shoot development via auxin signaling [35,36], which was supported by the morphological results (Figure 1).

## 3. Discussion

The effects of Gd on *S. rebaudiana* plant growth indicated significant toxicity, especially at high exposures of 1 and 3 mM. In a study on rice grown in hydroponic conditions, Liu et al. [27] found a low impact of 5 mM gadopentenic acid (Gd-DTPA) on rice growth and biomass, upon short-term exposure (up to 12 h), but higher impacts on rice plant growth for concentrations of 10 and 15 mM Gd-DTPA, which affected the rice growth. Another study on rice (hydroponics, 0–1 mM Gd exposure) showed significant inhibition of root growth for 0.1 and 1 mM [28]. The DW biomass of roots and shoots of rice was significantly decreased by 1 mM Gd, while the same concentration of 1 mM Gd led to severe decreases in root length (by 90.8%) and shoot length (by 58.5%). Gd-induced toxicity was also identified in a study on maize (hydroponics, 0–10 mg/L, 0–63 µM Gd/L) [32], in which 10 mg/L Gd was found to reduce shoot and root biomass by 67% and 37%, respectively. In a study on *Arabidopsis thaliana* [31] (hydroponics, 0–0.2 mM) Liu et al. found that at the lowest doses of 0.01 mM, Gd stimulated plant growth, while the 0.2 mM concentrations resulted in significant reduction in biomass and root length, by 15% and 50%, respectively, compared to controls.

Stimulation of chlorophylls synthesis occurred up to exposures of 0.1 mM GB, while the highest exposures of 1 mM and 3 mM led to a significant drop in chlorophyll synthesis. We attributed the stimulation of chlorophylls synthesis up to 0. 1 mM exposures to a mechanism of stress response of the *S. rebaudiana* plant. Additionally, plants normally synthesize higher amounts of chlorophyll A than B (needed for plant survival). In stress conditions, during the processes of chlorophyll degradation, Chl B may be converted into Chl A, leading to increased values of the Chl A/B ratio. Such was the present case where an increase in the chlorophyll a/b ratio from 2.23 to 3.01 (0.1 mM), 2.53 (1 mM) and 2.56 (3 mM) was observed. Liu et al. [31] found similar effects of decreasing the chlorophyll content by 16.7% upon exposure to 0.2 mM Gd, while at very low doses (up to 0.05 mM), no significant changes in chlorophyll synthesis were observed.

Carotenoids are important pigments in photosynthesis [37]. They are important as accessory light-harvesting pigments and as photoprotective pigments, protecting chlorophylls (among others) from oxidative stress. Basically, carotenoids and chlorophylls production followed the same trend upon increasing Gd exposures. Their drastic decrease for exposures to GB concentrations above 0.1 mM could be attributed to a failure of carotenoids to protect chlorophylls from oxidative stress. We observed a similar trend in a previous study on soybean plants exposed to ZnO nanoparticles [38].

Additionally, from the negative correlation between the ascorbic acid levels and the intensity of the free radicals resulting from the EPR, relevant information related to GB toxicity is obtained. On one hand, at low doses, decreases in the levels of free radicals are correlated with increased levels of AA, thus showing antioxidant plant defense, while at doses above 0.1 mM, even though the number of free radicals was increased, the synthesis of AA decreased, indicating plant dysfunctionalities. The strongly distorted Mn centers in the EPR spectra at high GB concentrations indicated plant dysfunctionalities, also confirmed by the levels of nutrients; for example, Zn.

Given all the results, it is expected that the photosynthesis in high-concentration Gd treated plants to be highly affected, which was confirmed by the symptomatic chlorosis for the last two samples, as shown in Table 1.

Already, from the lowest GB doses, variations in the concentrations of relevant metabolites were observed. The 0.1 mM exposure represents a threshold value. From the evolution of all analyzed parameters, it seemed that above 0.1 mM GB, the plant defense system became unable to cope with Gd-induced toxicity, resulting in severe inhibition of plant growth and production of relevant plant metabolites. It became obvious that above 0.1 mM, the stress generated by Gd treatment was too high and, consequentially, the plant defense system was not able to cope with it anymore, leading to morphological changes and drastic decreases in the carotenoids and chlorophylls levels. A previous study on tomato plants (hydroponic, 0–3 mM Gd exposure) [29] indicated that 0.5–0.7 mM was the tolerance level of Gd in tomatoes, while clear signs of toxicity were observed for 0.7 mM Gd.

## 4. Materials and Methods

### 4.1. Preparation of In Vitro Cultures, Plant Growth and Treatments

The basal medium for in vitro culture was Murashige and Skoog 1962 (MS) (Duchefa Biochemie B.V, The Netherlands, code M0222, including vitamins, original concentration 4405.19 mg/L). Commercial sugar was used as a carbon source (30 g/L). For medium gelification, 4 g/L Plant Agar (Duchefa Biochemie BV, The Netherlands) was used. The pH of the medium was adjusted to 5.8 before the agar was added. The culture vessels were 720 mL glass jars with a metal screw cap, fitted with a filter to ensure gas exchange. At first, 100 mL of culture medium was dispensed into each vessel, followed by addition of GB in final concentrations of 0.01, 0.02, 0.05, 0.1, 1 and 3 mM. Samples not exposed to gadolinium were used as controls. Next, the vessels containing the medium and the treatments were autoclaved for 20 min at 120 °C and 1 atm.

In order to set up the experiments, 30-day in vitro cultures of *S. rebaudiana* were used. The next day, after autoclaving the media, 20 nodal segments of *S. rebaudiana* were inoculated on each growth vessel. Incubation of plant cultures was performed in a growth chamber at a light intensity of 36 μmol × m^−2^ × s^−1^ (Philips cold white light fluorescent tubes, 36 W), a temperature of 23 ± 2 °C and 50 ± 5% humidity for 30 days.

For the quantifications of plant metabolites, ICP measurements and EPR analyses, freeze dried, powdered plant samples were used. More precisely, after 30 days of growth, the plants were carefully removed from the growth media, the roots were washed to remove remaining growth media, and then the plants were frozen to −80 °C and subsequently freeze dried (Telstar LyoQuest freeze dryer) and milled (MM40 RETSCH mill, Landsberger, Berlin, Germany). The powdered samples were stored at −20 °C prior to measurements. Each measurement was performed in triplicate. Each measurement was carried out on a mixture of powdered dried plants, with each mixture containing 30 plants.

### 4.2. Extraction of Chlorophylls, Carotenoids, and Ascorbic Acids

The carotenoids and chlorophylls were extracted using a mixture of petroleum ether:methanol:ethyl acetate (1:1:1, *v*:*v*:*v*). Briefly, 1 g of fresh weight *S. rebaudiana* sample was homogenized by grounding and exposed to 10 mL solvent, followed by homogenization, sonication of 15 min, and centrifugation at 11,000 rpm and 4 °C for 10 min [39]. Further, the supernatant was collected, and the remaining pellet was subjected once more to the extraction protocol described above. The procedure was repeated until complete extraction. The supernatants collected after each extraction step were combined together and washed three times with NaCl 15% in a separating funnel (1:1, *v*:*v*). At the end, the organic phase was collected and passed over anhydrous sodium sulfate to remove any traces of water. Subsequently, the organic phase was evaporated under vacuum at 35 °C, using a rotary evaporator (Heidolph Instruments GmbH & Co., Schwabach, Germany). The dried residue was blown with nitrogen and stored at −20 °C before HPLC analysis. To perform the HPLC analysis, the dried residue was redissolved in 1 mL of ethyl acetate.

Ascorbic and dehydroascorbic acids were extracted using an aqueous solution containing 3% metaphosphoric acid and 8% acetic acid. Next, 1 g of fresh weight, grounded *S. rebaudiana* sample was exposed to 4 mL aqueous solution, followed by thorough shaking for 30 s, sonication for 15 min, and centrifugation at 8000 rpm at 4 °C for 10 min. The supernatant was collected and passed through a 0.45 µm nylon filter prior to injection into the HPLC system.

### 4.3. HPLC Analysis

The extracted carotenoids, chlorophylls, and ascorbic acids were identified and quantified using an Agilent 1200 HPLC system, equipped with a solvent degasser, quaternary pumps, a DAD detector, and an automatic injector (Agilent Technologies, Santa Clara, CA, USA). The separation was performed on an EC 250/4.6 Nucleodur 300-5 C18 ec. Column (250 × 4.6 mm, 5 µm), at a temperature of 250 °C (Macherey-Nagel GmbH & Co. KG, Germany).

The mobile phase for carotenoids and chlorophylls analysis consisted of mixtures of acetonitrile: water (9:1, *v*/*v*) with 0.25% triethylamine (A) and ethyl acetate with 0.25% triethylamine (B) in the following gradient: at min 0, 0.90% A; at min 10, 50% A. The percentage of solvent A decreased from 50% at min 10 to 10% at min 20. The flow rate was 1 mL/min and the chromatograms were recorded at the wavelength λ = 410, 450 nm [40]. The HPLC peaks were identified and quantified using carotenoid and chlorophyll standards (Sigma Chemical Co. Ltd., St. Louis, MO, USA).

For ascorbic and dehydroascorbic acids analysis, the mobile phase consisted of a mixture of water: acetonitrile (95:5, *v*/*v*), with 1% formic acid. The flow rate was adjusted to 0.5 mL/min. Spectral values were recorded in the 200–400 nm range for all peaks. The chromatograms were recorded at the wavelength of 260 nm. For identification and quantification, L-ascorbic acid standard was used (Sigma Chemical Co. Ltd., St. Louis, MO, USA). For the MS detection, the positive ESI ionization mode was used, with 3000 V capillary voltage, 35 °C temperature, 8 L/min nitrogen flow, full scan, *m*/*z* 100–600.

Data acquisition and interpretation of results were performed using the Agilent ChemStation software.

### 4.4. Gd Influence on Plant Growth and Morphology

After 30 days of plant growth, the following parameters were measured in order to quantify potential effects of Gd on the *S. rebaudiana* growth: fresh and dry biomass, and root and shoot length. All indicated parameters were measured for at least 30 individual plants. The results obtained are expressed as mean ± SE.

### 4.5. ICP-OES

Quantification of NP uptake by the vegetal tissue was carried out using inductively coupled plasma optical emission spectroscopy (ICP-OES) using a Perkin Elmer—OPTIMA 2100 DV spectrometer (Perkin Elmer, Llantrisant, UK). Sample digestion was carried out as follows: the plant samples (dried and milled), about 0.2 g were solubilized using a strong oxidizing acid mixture (9 mL HNO_3_—65%, 1 mL H_2_O_2_—30%) for 16 h in a laminar flow hood. Subsequently, the same amounts as above of acid oxidizing mixture were again added to samples, followed by heating in a sand bath, and gentle boiling for two hours under reflux. At the end, the completely dissolved samples were quantitatively adjusted to 50 mL with slightly acidified water. The Gd uptake, as well as the Ca, Cd, Cr, Cu, Fe, K, Mg, Mn, Na, Pb, and Zn content, was quantified for control and Gd exposed samples. Calibration curves were recorded using Gd standard solution (1000 mg/L Gd, Merck). The other elements were quantified using a multielement standard solution IV (1000 mg/L, 23 elements, Merck).

### 4.6. MDA Assay

First, 0.1 g of powder dried plants were mixed with 3 mL of 0.1% trichloroacetic acid (TCA), mixed for 30 s followed by centrifugation at 10,000× *g*, 4 °C for 10 min. An aliquot of 0.5 mL supernatant was transferred into another tube and mixed with 1.5 mL of 0.5% thiobarbituric acid (TBA) prepared in 20% TCA. The samples were incubated for 30 min at 95 °C in a constant temperature water bath and then cooled in an ice bath for 5 min. After centrifuging at 10,000× *g* for 10 min at 4 °C, the absorbance of the obtained supernatant was detected at 532, 600, and 450 nm using a Lambda 25 (PerkinElmer Singapore) UV-Vis spectrophotometer. Lipid peroxidation was expressed as a sum of absorbance of polyunsaturated fatty acids (532 nm) and monounsaturated fatty acids (450 nm), respectively.

### 4.7. Electron Paramagnetic Spectroscopy (EPR)

For the EPR experiments, a known quantity (12–16 mg) of dried powdered plant was transferred to a Hirschmann 100 μL disposable glass capillary and placed in the Bruker EPR spectrometer, inside a Wilmad 3.8 mm × 241 mm (D × L) quartz tube as a guiding tube. The measurements recorded at room temperature were performed with an X-band EPR spectrometer with parameters as follows: 2500 G field center, 4800 G sweep width, modulation amplitude 5 G, sweep time 59.4 s, 9.46 mW power. Each spectrum was registered as an average of two scans. For each sample, four measurements were recorded.

### 4.8. Statistical Analysis

Ascorbic acids, carotenoids, and chlorophylls concentrations are presented as means ± standard deviation (SD). The variation of the ascorbic acids, carotenoids, chlorophylls concentration between controls plants and plants treated with 0.01, 0.05, 0.1, 1, and 3 mM GB were compared by analysis of variance (ANOVA) test and Dunnett’s multiple comparisons test using the GraphPad Prism^®^software package (version 8.01, GraphPad Software Inc., San Diego, CA, USA). A *p* < 0.05 value was considered statistically significant (* for *p*-value range 0.01–0.05; ** for *p*-value range 0.001–0.01; *** for *p*-value range 0.0001–0.01; **** for *p*-value range <0.0001).

## 5. Conclusions

The results of the present study showed that, at the highest doses (1 mM and 3 mM), GB exposure had detrimental effects, leading to significant inhibition in plant growth and in the concentrations of relevant compounds. Consequentially, the synthesis of carotenoids, chlorophylls A and B, was severely inhibited by these high exposures. For the highest dose of 3 mM GB, the Chl A and B concentrations decreased by 52.76% and 58.92%, and the levels of lutein, zeaxanthin, and beta-carotene were inhibited by 38.16%, 54.34%, and 54.34%, respectively. Additionally, at these high GB exposures, a significant decrease in concentrations of metal ions such as Cu, Zn, Ca, Fe, K, and Mn was recorded. A detrimental effect of the high GB doses was also confirmed through the increase in free radical species detected by EPR, which was negatively correlated with the evolution of antioxidant ascorbic acid, as shown in the discussion section. Given all these findings, the photosynthesis in high concentration Gd was highly affected.

The 0.1 mM dose of GB seemed to be a threshold, below which the plant defense system is functional, and the GB toxicity seemed to be low or moderate.

## Figures and Tables

**Figure 1 ijms-23-11368-f001:**
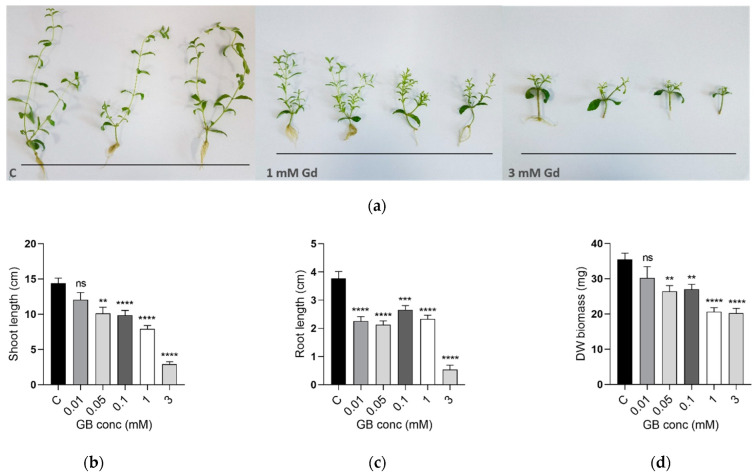
**Top**: (**a**) *S. rebaudiana* plants after being grown for 28 days in control growth media and media exposed to 1 mM and 3 mM gadobutrol (from **left** to **right**); **Bottom**: (**b**) mean and SE shoot length (*n* = 30), (**c**) mean and SE root length (*n* = 30) and (**d**) mean and SE dry weight plant biomass (*n* = 30) of *S. rebaudiana* plants after being exposed for 28 days to different concentrations of gadobutrol of 0–3 mM. (ns for *p*-value range > 0.05; ** for *p*-value range 0.001–0.01; *** for *p*-value range 0.0001–0.01; **** for *p*-value range <0.0001).

**Figure 2 ijms-23-11368-f002:**
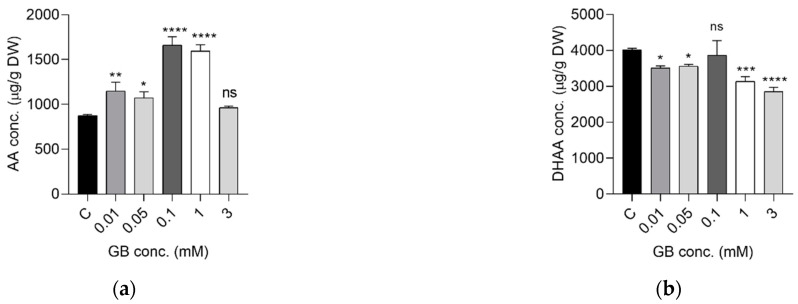
Ascorbic (**a**) and dehydroascorbic acids (**b**) concentrations in *S. rebaudiana* plants exposed to GB in concentrations of 0–3 mM. The concentrations are represented as mean and SD. (ns for *p*-value range > 0.05; * for *p*-value range 0.01–0.05; ** for *p*-value range 0.001–0.01; *** for *p*-value range 0.0001–0.01; **** for p-value range <0.0001).

**Figure 3 ijms-23-11368-f003:**
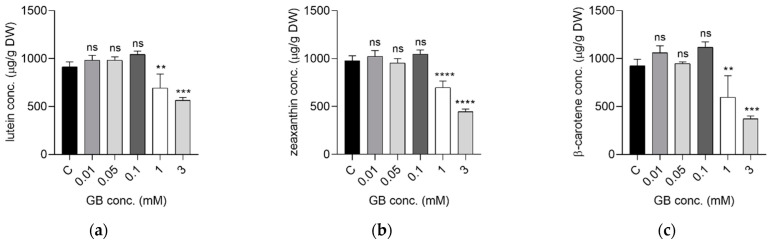
Concentration of carotenoids in *S. rebaudiana* plants treated with GB in concentrations 0–3 mM; lutein (**a**), zeaxanthin (**b**), β-carotene (**c**) (ns for *p*-value range > 0.05; ** for *p*-value range 0.001–0.01; *** for *p*-value range 0.0001–0.01; **** for *p*-value range <0.0001).

**Figure 4 ijms-23-11368-f004:**
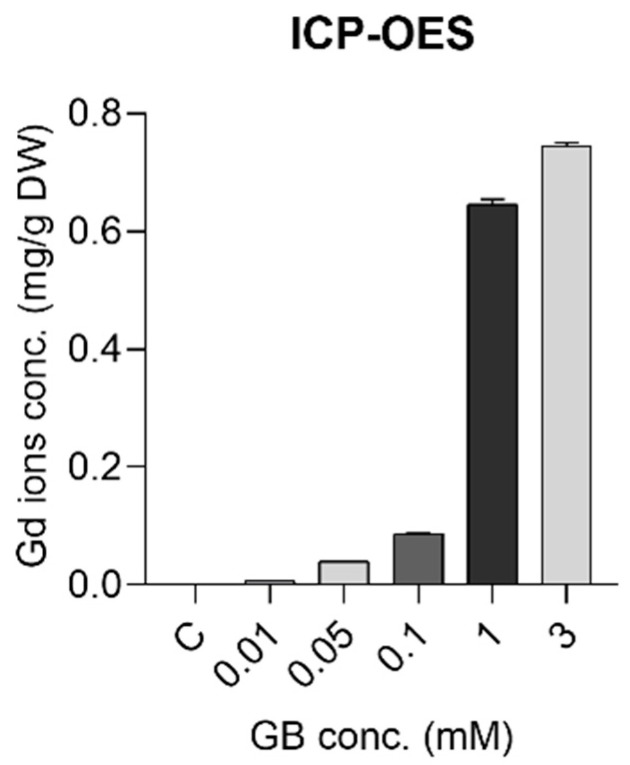
Uptake of Gd ions in the *S. rebaudiana* plant tissues exposed to GB in concentrations 0–3 mM, quantified by ICP-OES. The results are presented as mean and SD.

**Figure 5 ijms-23-11368-f005:**
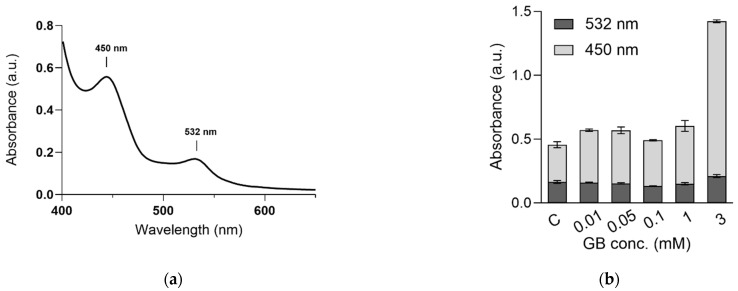
Characteristic UV-Vis spectra of the compounds resulted from the MDA assay (**a**) and the evolution of the absorbances of the 450 and 532 nm peaks with the increase in the Gd exposure (**b**).

**Figure 6 ijms-23-11368-f006:**
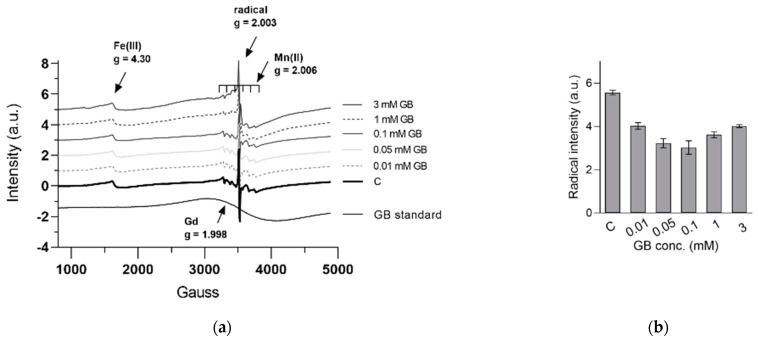
EPR spectra of the *S. rebaudiana* plants exposed to GB in concentration 0–3 mM and a Gd standard at 330 µg in capillary, at room temperature, 9.46 mW power. The *g*-values of the signals are indicated (**a**); radical intensity of the *S. rebaudiana* plants exposed to GB measured by peak-to-peak difference at I_3513G_–I_3527G_ (**b**).

**Table 1 ijms-23-11368-t001:** Variations in chlorophylls a (Chl A) and b (Chl B) upon exposure to GB in concentrations up to 3 mM. The quantifications were carried out using HPLC-DAD. The results are given as mean ± SD.

GB Concentration (mM)	Chl A (µg/g DW)	Chl B (µg/g DW)	Chl A/B
0	5717 ± 529	2560 ± 49	2.23
0.01	8447 ± 314	2986 ± 83	2.82
0.05	8269 ± 268	2898 ± 95	2.85
0.1	8705 ± 296	2887 ± 61	3.01
1	5064 ± 243	1998 ± 110	2.53
3	2700 ± 284	1051 ± 38	2.56

**Table 2 ijms-23-11368-t002:** Concentrations of minerals in dried, powdered samples of *S. rebaudiana* after exposure to 0–3 mM GB.

Element	GB Concentration (mM)					
	0	0.01	0.05	0.1	1	3
Ca (mg/g)	2.681 ± 0.070	2.784 ± 0.040	2.812 ± 0.018	2.758 ± 0.028	2.622 ± 0.081	1.787 ± 0.052
Cd (mg/g)	0.000 ± 0.000	0.000 ± 0.000	0.000 ± 0.000	0.000 ± 0.000	0.000 ± 0.000	0.000 ± 0.000
Cr (mg/g)	0.002 ± 0.001	0.001 ± 0.001	0.001 ± 0.001	0.001 ± 0.001	0.001 ± 0.001	0.003 ± 0.001
Cu (mg/g)	0.011 ± 0.001	0.012 ± 0.001	0.008 ± 0.001	0.008 ± 0.001	0.000 ± 0.000	0.000 ± 0.000
Fe (mg/g)	0.199 ± 0.002	0.207 ± 0.003	0.215 ± 0.002	0.209 ± 0.002	0.230 ± 0.003	0.141 ± 0.001
K (mg/g)	30.019 ± 0.237	34.737 ± 0.320	35.746 ± 0.351	37.906 ± 0.201	34.293 ± 0.292	23.471 ± 0.129
Mg (mg/g)	1.149 ± 0.001	1.260 ± 0.002	1.315 ± 0.005	1.286 ± 0.016	1.281 ± 0.008	1.834 ± 0.018
Mn (mg/g)	0.102 ± 0.001	0.107 ± 0.002	0.104 ± 0.002	0.106 ± 0.002	0.095 ± 0.003	0.064 ± 0.006
Na (mg/g)	0.264 ± 0.002	0.340 ± 0.002	0.412 ± 0.003	0.466 ± 0.006	0.328 ± 0.002	0.272 ± 0.003
Pb (mg/g)	0.003 ± 0.001	0.002 ± 0.001	0.003 ± 0.001	0.001 ± 0.001	0.002 ± 0.001	0.000 ± 0.000
Zn (mg/g)	0.178 ± 0.006	0.094 ± 0.003	0.094 ± 0.002	0.093 ± 0.003	0.108 ± 0.001	0.08 ± 0.001

## Data Availability

Not applicable.

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
