# Peer review of "Gadolinium Accumulation and Toxicity on In Vitro Grown Stevia rebaudiana: A Case-Study on Gadobutrol"

_ijms, 2022, doi:10.3390/ijms231911368_

Round 1
Reviewer 1 Report
Please see the attached file about the review comments.

Author Response
Please, see the attachment.

Reviewer 2 Report
The Authors investigated an interesting topic: how the gadolinium pollution can impact to environment and living being. Gadolinium is a metal present in several contrast agents and the medical investigations contribute to its diffusion in the environment. For their study the Authors investigated how the treatment with different concentrations of Gadobutrol (a contrast agent) affect morphology, metabolites, chlorophylls, elements content and oxidative stress in Stevia rebaudiana., a plant very useful for the production of sweeteners(widely used by diabetics). The study design is good, the experiments have been well realized, The results are clear, the conclusion are appropriate and also the references are appropriate. In my opinion the paper is interesting and could deserve the pubblication.
Only some indications:
please, pay attention to the typos, i.e. line 339 obsrved, line 360 difficut, line 365 indicationg, line 425 symthomatic
please rename section 3 as Results, in the manuscript is results and discussion but the section 4 is Discussion
I suggest to the Authors want to write in the conclusion how they will continue their study on this interesting and current topic
Author Response
Plase, see the attachment.

Round 2
Reviewer 1 Report
I am happy with the authors' responses and the improvement of manuscript. I suggest it could be accepted for publication.